# Goal-line oracles: Exploring accuracy of wisdom of the crowd for football predictions

**Jens Koed Madsen**　*

Department of Psychological and Behavioural Sciences, London School of Economics and Political Science, London, United Kingdom

* j.madsen2@lse.ac.uk

**Data Availability Statement:** The minimal data set associated with this manuscript is publicly available via the Open Science Framework repository. Data can be accessed via the following URL: https://osf.io/kg5rv/.

## Abstract

Football's inherent volatility and low-scoring nature present unique challenges for predicting outcomes. This study investigates the efficacy of Wisdom of the Crowd in forecasting football match outcomes as well as expected goals (XG) across a Premier League season. Participants predicted team goal counts, which were then compared to actual expected goals (XG) and match results. Results across 760 team predictions reveal that while Wisdom of the Crowd accurately predicts XG on average, it overestimates 'big-6' teams and underestimates others, hinting at inherent biases. Notably, however, collective crowd predictions outperform individual estimates consistently, affirming the power of collective intelligence. Furthermore, when tested against betting odds, Wisdom of the Crowd demonstrates limited profitability, indicating its potential as a supplementary rather than sole prediction tool. These findings underscore the nuanced dynamics of football prediction and highlight the utility of collective wisdom in navigating its complexities. Overall, this study contributes insights into crowd prediction dynamics and underscores its potential in football analytics, shedding light on its implications for decision-making and understanding group behaviour in sports forecasting contexts.

## Introduction

Association Football (hereafter referred to as 'football') is a beautiful game that is notoriously difficult to predict. It is a low-scoring game, which means that the occurrence of a single goal can drastically change how the rest of the match unfolds, as teams may change their strategy after a goal. For example, if the favourite scores early, they may decide to sit back and rely on their defence to see out the game. However, this makes them vulnerable to a last-minute equaliser, which is subsequently hard to rectify. As such, results, as well as strategy in football are very volatile, as minor changes to scores can change them fundamentally, even early in a match. Comparatively, in high-scoring sports like basketball or handball, going behind by a few points is not necessarily strategy-changing unless it happens toward the end of the match. This is because favourites may believe they will close the gap throughout the game.

Football's dynamic nature complicates statistical analyses. For comparison, baseball is a more static game where identical game situations occur with high frequency. The pitcher is

**Funding:** The author(s) received no specific funding for this work.

always located at the same distance to the batsman, which makes it easier to compare and statistically analyse the outcome of different pitch types or strategies. Football situations, on the other hand, are incredibly dynamic and rarely replicate exactly–a shot may be taken from a slightly different angle, a defender may be placed differently in relation to the sight of the goal, and so forth. Thus, while a shot may occur from similar positions, multiple factors influence the probability of scoring a goal. This makes it harder to compare the success rate of different types of shots, as in-game situations seldom replicate exactly. Given a dynamic and volatile game, predictions are difficult.

Due to the dynamic and low-scoring nature of the game, predicting the outcome of games is hard. Indeed, favourites are less likely to win in football compared with other major sports such as handball, basketball, American football, and baseball, [1, p. 53]. Predictions are made more difficult by the fact that football matches have three possible outcomes (win/lose/draw) rather than two (win/lose), making random guesses less likely to succeed. People have used different approaches to predict football results. These methods include structural Bayesian models [2,3], logistic regression modelling [4], and machine learning [5]. Common to these endeavours is an attempt to build accurate prediction machinery based on prior results and outcomes to predict future results. Machine Learning models may yield predictive successes of around 65%, with a potential for some improvements [5].

Alongside computational models trained on old results to predict future matches, a different and complementary area of interest is whether people can predict football results. That is, testing the insights of laypeople or experts in predicting the outcome of matches. This realm touches on aspects like decision-making and psychological biases. Foundationally, it asks if people entertain accurate perceptions of the world. In this paper, we present data from a Wisdom of the Crowd study that runs over an entire Premier League season. Throughout the 2022/23 Premier League season (38 rounds), participants guess the score of each match (10 matches in each round), which can be compared with the actual score as well as the Expected Goals for each team. With two teams competing in each match, we have Wisdom of the Crowd predictions for 760 team performances across a season that we may compare with actual match outcomes.

## Literature review

*Wisdom of the Crowd* is a phenomenon where the collective judgment of a group of individuals is assumed to have a more accurate judgment of an event or result compared with any single member of the group, including experts. This concept suggests that when a group of people independently provides their opinions or estimates on a particular issue or question, the average of these responses may outperform the predictions of individual members. Factors such as the independence of individual judgments, the decentralization of decision-making, and the aggregation of diverse opinions may influence such effectiveness of the Wisdom of the Crowd. Additionally, the larger the group, the more accurate the collective judgment is assumed to be, as it reduces the impact of outliers and random errors. This method of prediction has been applied to areas such as politics [6,7].

Of relevance to the current study, Wisdom of the Crowd has been used for football predictions [8]. In this paper, the rank-order choice paradigm, adapted from the *collective recognition heuristic* [9], is used to predict the outcome of tennis and football matches. They find that crowd rankings are more accurate than aggregated betting odds and official ranking predictions. Similarly, [10] looks at methods of extracting Wisdom of the Crowd predictions for the 2018 World Cup where they test different aggregation strategies while [11] describes how text mining of social media can be used to set crowd predictions. Our study differs from past

crowd prediction studies in important ways. First, we provide predictions over a season of 38 Premier League rounds (a total of 380 matches), which provides measures across a whole year. Second, we compare crowd predictions to several metrics, including actual goals scores as well as XG (see below). As such, we add to the literature on crowd predictions for football. Related to the current study, other papers explore crowd predictions and transfer market valuations (e.g. [12–14], see [15, 16] for a review of the transfer market calculation and how it can be used in football). For example, [12] shows that crowd evaluations are more accurate than FIFA or ELO rankings (see also [17] for crowd valuation for player investments). FIFA rankings are the official country and club rankings by the *Fédération Internationale de Football Association*. ELO is a method of calculating relative skill levels in zero-sum games such as chess, football, and computer games. The difference in ELO ratings between two players yields a probabilistic predictor of the outcome of a match.

In football, the outcome of a match may not be indicative of the team's performance or dominance. It is entirely possible for one team to dominate possession and chances and yet to lose if the opposing team breaks away and scores a counterattack or from a freekick. The Expected Goals (XG) metric is used as an alternative measure to gauge the quality of chances that a team produces over a game [18]. This is a metric that calculates the probability of scoring from a given chance [19]. For example, around 76% of penalties result in a goal while a shot from afar with defenders in the way may only result in a goal 3–5% of the time that this is attempted. XG for a given chance indicates the average probability of that type of chance. As such, if a team is awarded a penalty, their XG score is + 0.76. Over the game, a team accumulates XG per shot they take. This is meant to aggregate the probability of scoring across the chances created. Thus, a large XG of 2.87 would indicate that this team, on average, should expect to score between 2 and 3 goals given the quality of chances they created. Conversely, if a team only generates an XG of 0.12, they were unlikely to score in that game given the chances they created. Of course, the team may over- or underperform in relation to their XG. A team may score 2 goals from very low probability chances or score none despite dominance and the creation of big chances. While most players (and teams) should regress to the mean across the season, exceptional players may outperform their personal XG, as they are better at shooting than average players. Of course, there are limits to XG as an accurate measure of dominance and chance-creation. For one, it is difficult to calculate the exact probability of converting a chance to a goal given the fact that most situations are somewhat unique (e.g., distance from goal may be similar in two situations while the placement of defenders and supporting attacking players may increase or decrease the probability in different situations). Additionally, there are different ways to measure chance creation and the probability of conversion. For example, Spearman [20] uses off-the-ball metrics (e.g., a tall player who is unmarked during a corner kick). In this paper, 'predicted XG' refers to the Wisdom of the Crowds prediction for a given team while 'observed XG' refers to the XG reported after the match on Understat, which is a public website that tracks Expected Goals across several European leagues.

To capture dominance as well as results, we compare Wisdom of the Crowd guesses of the number of goals per team per match to XG as well as guesses of the result of each match to the actual match outcome. This indicates whether participants can predict results as well as predict the quality of chances that a team is expected to produce, regardless of whether this results in victory. As such, we have two measures for comparison with different benefits and limitations. First, the Wisdom of the Crowd predictions compared with actual goals scored benefit from the fact that goals are observable and genuine outcomes. However, given the considerable noise in football outcomes, this is subject to variance. For example, a team may be dominated by their opposition, produce very few chances, and still win 1–0 on an extremely unlikely shot from far away at the end of the game. In this case, predictions may be wrong on the outcome

(the dominated team winning) but right in predicting the level of domination. Second, Wisdom of the Crowd predictions compared with XG benefit from comparing expected dominance (predictions) with actual dominance (XG expressed as the number and quality of chances a team produces). However, this is limited by the fact that XG does not necessarily translate into actual outcomes. By comparing predictions to both, we get a richer picture of the Wisdom of the Crowd's capacity to predict football matches (outcome or chance-creation).

We provide guesses for each team involved in each Premier League game across the 38 rounds. This means we have 760 team estimates across the season. Given the past successes of Wisdom of the Crowd to outperform individuals and estimate complicated issues, we proposed the following hypotheses.

H1: Wisdom of the Crowds XG estimates will not differ from observed XG.

H2: Wisdom of the Crowds estimates of match results will outperform estimates from individual participants throughout the season.

## Method

We want to elicit people's predictions on the number of goals they expect from each team for each game to be able to generate predictions and compare them with XG and the actual outcome of the game. Participants received a Qualtrics link for each round. When opening the link, participants were asked for their consent. Data was collected over the 2022/2023 season with a weekly guess for each round from August 2022 to May 2023. For each game in each round, participants saw the combination of teams playing each other (with the home team on the left and the away team on the right). For example, participants would be asked 'How many goals will Crystal Palace score against Arsenal FC?' and 'How many goals will Arsenal FC score against Crystal Palace?' Participants could indicate any number in a box (typically, between 0 and 5). Each participant thus provides evaluations of XG as well as the result of the game. We average these guesses to estimate the Wisdom of the Crowd prediction for each team for each game.

## Participants

Participants were recruited opportunistically. Every week, we emailed people who had listed their interest in the project as well as distributed the link to the survey on social media platforms like LinkedIn and X (then, Twitter). As this was a project among colleagues and friends with the added recruitment through social media, no payment was offered to take part. Each participant was given a random ID that they could enter when providing their estimates to be able to track individuals across all 38 match rounds. We use this ID to compare outcome predictions for individuals who take part in each round and the predictions of the Wisdom of the Crowd.

The opportunistic nature of the sampling has two limitations. First, the number of participants varied from round to round (the lowest participation during a holiday was 15 while the highest number of participants in a round was 85). The average number of participants for each round was 25.21. To gauge if the number of participants is problematic for analyses, we compare rounds with 25 or fewer participants with rounds with more than 25 participants. We see no significant difference in observed XG between these groups (t = 1.559, p = 0.119) or between predicted XG (t = 1.755, p = 0.079). Additionally, we observe similar correlations between predictions and observations ($R^2$ = 0.205 for rounds with fewer than 25 participants;

$R^2$ = 0.168 for rounds with more than 25 participants). As such, it appears that there are few structural differences between rounds with fewer or more than 25 participants.

For each round of matches, we average the scores of each participant to generate the prediction for expected goals, which is then compared with XG. For each round, there are ten matches (i.e., ten options for comparing predictions to observations). As there are 38 rounds, we have 380 averages in total to be compared with observations. Given the study design, we run a *G*Power 3.1* [21] analysis of repeated measures (within factors) for an ANOVA to determine the sample size needed to minimise type-I and type-II errors and ensure a strong statistical power. Using *effect size (f)* = 0.2, *significance level (α)* = 0.05, *power level (1-β)* = 0.95, number of groups = 1, number of measurements = 38, and a *nonsphericity correction* = 0.5, a sample size of 21 participants is recommended. As the study averages 25.21 participants per round, it suggests that the study was adequately powered. Nonetheless, some rounds have fewer than required participants, making them more vulnerable to variation or biases from individual respondents. However, as discussed below, when looking at the percentage of correct guesses in weeks with fewer than 20 participants, there are no differences compared with rounds that are adequately powered. Therefore, with some hesitation concerning some rounds, the study appears reasonably powered. The results should nonetheless be seen in this light and future research should replicate the study with more respondents for all rounds.

Second, the nature of opportunistic sampling meant that participants were likely to be people who have an interest in football. This means that participants were most likely more interested in football than average citizens. Additionally, participants may be fans of a particular club, which may skew their perceptions of their preferred club. For example, an ardent Arsenal fan may have biased views of their performance. Given the relatively small number of participants, this makes the results vulnerable to individual biases such as club allegiance.

## Results

In the following, we report results from the Wisdom of the Crowd study with regard to expected goals and actual outcomes across a whole Premier League season (380 matches).

### Comparing WoC estimates with expected goals

To compare the Crowd's predicted XG with the observed XG, we take data from Understat. This is a source that has been used previously to gauge metrics from football results [22]. The Crowd's predicted XG per match per team ($\mu$ = 1.43, $\sigma$ = 0.66) appears close to the observed XG per team per match ($\mu$ = 1.48, $\sigma$ = 0.89). An independent two-tailed t-test shows no significant difference between predicted and observed XGs per team per match (t = 1.328, p = 0.1844). This suggests that Crowd participants could reasonably predict the quality of chances created by each team across all matches.

However, individual matches have significant variation, as some teams create many chances while other teams are less able to do so. Further, some teams may have a stronger season than predicted if the team does exceptionally well–or vice versa, some teams may underperform across a season. Analysing predictions by team reveals interesting psychological heuristics. First, the mean variation of observed XG (0.79) is considerably higher than the Crowd's predicted variation (0.33). Using an F-test, the difference in variation is significant (F = 0.54, p, 0.001). This suggests football matches are more chaotic than people in the Crowd may believe and that individual teams vary more from game to game than people may believe. This is in line with for example the observation that favourites are less likely to be victorious in football [1]–given greater deviation in performance from game to game, it increases the probability that favourites lose or draw.

**Table 1. Team differences in XG between prediction and observation.**

| Team | Predicted XG | Actual XG | XG difference |
|---|---|---|---|
| 1. Brighton | 1.37 (0.37) | 2.04 (0.83) | 0.67 |
| 2. Newcastle | 1.64 (0.38) | 2.02 (1.13) | 0.38 |
| 3. Everton | 0.94 (0.27) | 1.28 (0.75) | 0.34 |
| 4. Bournemouth | 0.79 (0.24) | 1.05 (0.68) | 0.26 |
| 5. Brentford | 1.29 (0.33) | 1.55 (0.85) | 0.26 |
| 6. Nottingham Forest | 0.79 (0.29) | 1.04 (0.62) | 0.25 |
| 7. Southampton | 0.84 (0.28) | 1.03 (0.67) | 0.20 |
| 8. Leeds | 1.10 (0.31) | 1.26 (0.66) | 0.16 |
| 9. Fulham | 1.11 (0.32) | 1.26 (0.73) | 0.15 |
| 10. Leicester | 1.16 (0.40) | 1.29 (0.83) | 0.13 |
| 11. Aston Villa | 1.23 (0.37) | 1.31 (0.84) | 0.08 |
| 12. West Ham | 1.27 (0.32) | 1.35 (0.83) | 0.08 |
| 13. Manchester United | 1.88 (0.38) | 1.89 (1.09) | 0.01 |
| 14. Wolverhampton | 0.94 (0.24) | 0.92 (0.53) | - 0.02 |
| 15. Liverpool | 2.25 (0.45) | 2.13 (1.02) | - 0.12 |
| 16. Crystal Palace | 1.23 (0.33) | 1.07 (0.71) | - 0.16 |
| 17. Tottenham | 1.73 (0.36) | 1.52 (0.59) | - 0.20 |
| 18. Chelsea | 1.65 (0.36) | 1.36 (0.63) | - 0.28 |
| 19. Arsenal | 2.44 (0.29) | 2.01 (0.86) | - 0.43 |
| 20. Manchester City | 2.91 (0.48) | 2.22 (0.92) | - 0.69 |

Table 1 describes the predicted and observed XG by team. As it appears, Wisdom of the Crowd in some instances under-estimates the actual XG while the Crowd over-estimates for other teams. As discussed below, this appears linked to a bias to overestimate perceived favourites.

The traditional 'big-6' teams (Arsenal, Chelsea, Liverpool, Manchester City, Manchester United, and Tottenham) all appear toward the bottom of this table. This suggests that people had a biased heuristic of overestimating their ability to create chances. Averaging across the big six teams, we observe a difference between predicted and observed XG of 0.28, suggesting that people believe these teams should create more chances than they do. In line with this, promoted teams are under-estimated by 0.22 and other teams are under-estimated by 0.19. Promoted teams are the ones who moved from a lower league (Championship) to the Premier League and may thus be perceived to be less strong or dominant compared with established Premier League teams.

Beyond averaging, we can test whether participants can predict the XG of each team for each game. Using a linear regression model, Wisdom of the Crowds has a statistically significant positive correlation ($R^2$ = 0.1945, $p < 0.001$, see Fig 1).

In sum, the Wisdom of the Crowd appears to somewhat accurately predict the XG of teams, as we observe no significant difference between the average raw Wisdom of the Crowds estimates and the observed XG (t = 1.328, p = 0.1844). To further probe the data, we conduct Mean Absolute Error (MAE) and Root Mean Square Error (RMSE) analyses of predicted and observed XG. This yields an MAE of 0.645 and an RMSE of 0.844. As MAE uses absolute values of errors rather than squaring errors before averaging as RMSE, the former is less sensitive to outliers. Given a typical range of XG of 0–4, the MAE and RMSE analyses suggest that model predictions are reasonably close to the observed XG values. However, there is definite

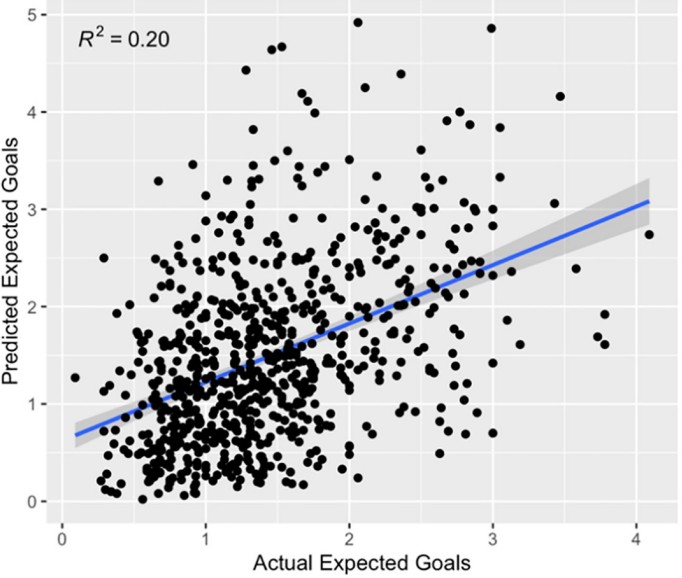

**Fig 1. Predictions and observations for each team for each game.**

room for improvement, which suggests that the correlations and closeness of averages should be taken with a grain of salt.

As we see a positive correlation between predictions and XG as well as observe no significant difference between estimates and overall mean XG per game, the results provide tentative support for H1 (Wisdom of the Crowds XG estimates will not differ from observed XG). These predictions positively correlate with observed XG. However, we note that the correlation is relatively weak (around 19.45% of the variance is explained by the model). Given the chaotic and dynamic nature of football, this is not surprising. As mentioned previously, matches can vary greatly in low-scoring games, as early goals may encourage otherwise dominant teams to sit back and consequently create fewer chances than expected throughout the match.

When exploring each team in detail, we observe a bias toward over-estimation of 'big-6' teams and under-estimation of the other teams. This indicates a psychological bias where favourites are believed to create more high-quality chances than they do. Further, we find that observed XG varies significantly more between games than participants expect (F = 0.54, p, 0.001). Along with the overestimation of favourites, this indicates that football matches are more volatile than people expect. However, when looking at each team's predicted versus observed XG, we see a strong positive correlation ($R^2$ = 0.1945). In sum, this suggests that the Wisdom of the Crowd is reasonably accurate, but that fundamental biases on favourites and stability may persist. We therefore find support for H1.

## Comparing WoC estimates with match outcomes

In addition to XG, we were interested in whether the Wisdom of the Crowd would outperform individual participants as well as betting agencies concerning the results of each game. For each round, we calculated the predicted result for each game (which team the Crowd collectively believed would win or if the game would be a draw). To calculate predicted outcomes, we round each prediction to the nearest whole number, such that an XG of 2.91 would predict 3 goals while 1.15 would predict 1 goal. However, this may yield counter-intuitive predictions.

For example, in a game where one team had a predicted XG of 1.51 and the other 1.49, simply rounding up and down would indicate a 2–1 victory, despite a predicted XG difference of 0.02. To counter this, a predictive heuristic was set to correct for such counter-intuitive situations. If XG predictions were within 0.2, we would predict a draw (e.g. 1.48–1.52 would be a draw). The same problem exists in the other direction. For example, in a game where one team had a predicted XG of 0.51 and the other 1.49, simply rounding up and down would indicate a 1–1 draw, despite a predicted XG difference of 0.98. To counter this, another predictive heuristic was set. If XG predictions were bigger than 0.5, we would predict a win to the team with the bigger XG prediction (e.g. 0.53–1.38 would be a win rather than a draw).

As described above, we used a heuristic to assist predictions (e.g., if the predicted gap between two teams is less than 0.2, we estimated a draw). To test the robustness of the 'draw' heuristic, we trial an alternative model where differences of 0.1 or less (instead of 0.2) yield a draw. That is, if Team A has an ExG of 1.23 and Team B has an ExG of 1.05, the original heuristic would predict a draw (as the difference is less than 0.18) while the new heuristic would predict that Team A would win. While the original heuristic yielded an overall prediction rate of 50% exactly, the new heuristic does slightly better and yields a prediction rate of 52.1%. This suggests that the original heuristic was set too wide and that a difference of 0.1 or less may be a better heuristic for estimating draws. Of course, as only 94 of the total 380 matches were predicted as draws, we cannot say if the differences are statistically significant over a larger sample size. Nonetheless, the results suggest that a tighter heuristic would be more successful as a prediction assistance.

To gauge the robustness of the predictions, we compare predictions with other methods of gauging the Wisdom of the Crowds, such as Navajas et al. [23] or taking the median rather than the mean estimate. Navajas and colleagues [23] provide an alternative method to generate WoC estimates. Here, participants provide individual estimates. After this, they are grouped in small groups to discuss the same question to reach a consensus. The final collective estimate is derived by aggregating these group consensus decisions, which is more accurate than averaging individual opinions. However, as we cannot get participants to deliberate, we provide a rough approximation of this method by dividing respondents into five random groups and taking the mean of these sub-groups (representing synthetic consensus). Unsurprisingly, the aggregate eventually yields more or less the same outcomes as just averaging across all participants (comparing the Navajas approximate with simple means via a linear regression yields an R2 of 0.998, indicating that the results are the same). We therefore do not pursue the Navajas method further.

As an alternative to using the mean, we may query if taking the median yields a better prediction. To do so, we take the median value of each prediction to generate the estimated number of goals for each team. However, this has worse overall predictive success than the averaging model, as the median model only yields 47.6% correct guesses (compared with 50% for the 0.2 averaging model and 52.1% for the 0.1 averaging model). Branching across the three models, we observe the following rate of success: Mean estimate (0.1 draw heuristic): 52.1% correct, Mean estimate (0.2 draw heuristic): 50.0% correct, Median estimates: 47.6% correct. This indicates that the best WoC method for the current project is aggregate means with a draw heuristic of 0.1.

## Comparing WoC estimates with individual performance

To compare Wisdom of the Crowd with individual performance, we use the predicted results for each round and track the percentage of games that the Wisdom of the Crowd correctly predicted and compared this to how well individuals predicted correctly in each of the 38 rounds.

**Table 2. Percentage correct across 38 rounds.**

| Predictor | Correct |
|---|:---:|
| Wisdom of the Crowds (0.1 heuristic) | **52.1%** |
| Wisdom of the Crowds (0.2 heuristic) | 50.0% |
| Player ID: 8806f8f2 | 48.1% |
| Wisdom of the Crowds (median heuristic) | 47.6% |
| Player ID: 82b6e5a5 | 47.4% |
| Player ID: 3a1ce2bf | 45.5% |
| Player ID: db459719 | 41.1% |

Below we report the result of this comparison for four participants who took part in all 38 rounds. Consequently, we compare their results to that of the Wisdom of the Crowd (see Table 2).

As evident, Wisdom of the Crowd outperforms all individual players across the 38 rounds. This suggests that collective predictions are better than individual predictions. In sum, we find support for H2. Unfortunately, only four players participated in more than 50% of the rounds (the threshold set for being compared with the model performance), which means the belief in the conclusion should be taken with some moderation. Future studies should investigate comparing Wisdom of the Crowd predictions against a larger number of participants.

To measure if Wisdom of the Crowd would beat the odds, we looked up betting odds for each game and kept track of outcomes. We pretended to bet £10 on each of the 10 matches per round (yielding a fictional investment of £3600). For example, in the Everton-Nottingham Forest game (Round 3), participants estimated that Everton would have an XG of 1.20 (actual, 1.43) while Nottingham would have an XG of 0.91 (actual 0.92). This would indicate a predicted 1–1 draw. We used an odds-tracked to glean the highest odds for this prediction. In this case, the odds for a draw were 13–5 which would yield a pay-out of £36 (i.e., a payment of £26 given the £10 investment). This allowed us to accumulate all gains and losses across the season. However, we tracked each prediction and their hypothetical payouts or losses across 36 rounds. The predictions from the Wisdom of the Crowd more-or-less broke even, as the hypothetical betting scheme lost £57.96 over the 36 matches (on the back of a hypothetical investment of £3600—we did not compare predictions with betting outcomes in the first two rounds as the idea only occurred to us from Round 3 and odds could not be recovered from Rounds 1–2). Of course, the percentage of correct games does not necessarily entail high rewards in gambling. For example, predicting that Manchester City would beat Leeds United at Home (Round 35) only yielded odds of 1/6, meaning that a £10 would pay £11.67 (meaning a pay-out of £1.67 after the cost of the bet was subtracted). However, this has a high probability of being correct. Comparatively, predicting a draw between Everton and Leicester (Round 15) is less likely to be correct and yields higher odds of 23/10, meaning a pay-out of £23 after the cost of the bet was subtracted). This suggests that the naïve Wisdom of the Crowd (simple aggregation of estimates) is *not* a viable method to beat the betting companies. This analysis differs from H2, as we did not compare what individual participants would have bet on–as such, the betting analysis is supplementary rather than substantial to H2.

## Discussion and concluding remarks

This study explores the predictive potential of Wisdom of the Crowd for football matches over an entire English Premier League season. Specifically, we demonstrate that Wisdom of the Crowd XG estimates positively correlate with observed XG (H1), that Wisdom of the Crowd

XG estimates did not differ from observed XG (H1), and that Wisdom of the Crowd estimates of match results outperform estimates from individual players throughout the season (H2). Our results support both hypotheses and suggest that Wisdom of the Crowd across a season can reasonably capture dominance and outcome of football matches, even though this is a difficult and dynamic game to predict.

While yielding interesting insights, the current study has several limitations. First, the predictions were only made for one Premier League season. This makes predictions vulnerable to team performances that under or overperform, which may not be replicated across several seasons. As an extreme example of this, Leicester City won the English Premier League in 2015/2016 despite being touted for relegation that season. Predicting this performance before and even during this season would be improbable and Leicester were eventually relegated in the 2022/2023 season, meaning that this level of performance was not sustained long-term. Future research should conduct larger studies across multiple seasons and leagues to examine the stability of Wisdom of the Crowd predictions. Second, given the opportunistic sampling, some weeks only had 16–17 participants, which made predictions more vulnerable to partisan predictions. For example, one continuous player–a staunch Arsenal supporter–consistently guessed that strong rival teams such as Tottenham, Manchester United, and Chelsea would lose 0–4 regardless of the opposition. Despite this bias, we included them in the analysis. In a larger sample, we expect more partisan outliers, which should balance out each other.

However, when looking at the percentage of correct guesses in weeks with fewer than 20 participants, there are no differences. As with the latter limitation, future research could run a study with 100+ participants each week to limit the impact of partisan predictions. On the other hand, it is worth noting that Wisdom of the Crowds performs well in predicting XG and actual match outcomes even in situations with 20–30 participants, which suggests that large numbers of estimators may not be needed. Finally, we do not collect demographic information. It would be interesting to explore if prior knowledge of football, age, or other characteristics make people more or less accurate. Indeed, the study raises questions on how to identify the right combination of people to form a crowd (see e.g. [24]) as well as the influence of social dynamics [25]. Targeted and larger crowds may outperform the current results.

In all, the results across the predictions suggest that Wisdom of the Crowd accurately predict average XG. The results indicate some inherent biases, such as overestimation of the 'big-6' teams. Notably, the collective crowd predictions consistently outperform individual estimates, affirming the power of collective intelligence. When we test predictions against betting odds, Wisdom of the Crowds has limited profitability (barely breaking even), indicating its potential as a supplementary rather than sole prediction tool. The results showcase the nuanced dynamics of football prediction. Overall, this study contributes insights into crowd prediction dynamics and underscores its potential in football analytics, shedding light on its implications for decision-making and understanding group behaviour in sports forecasting.

## Author Contributions

**Conceptualization:** Jens Koed Madsen.

**Data curation:** Jens Koed Madsen.

**Formal analysis:** Jens Koed Madsen.

**Methodology:** Jens Koed Madsen.

**Visualization:** Jens Koed Madsen.

**Writing – original draft:** Jens Koed Madsen.

**Writing – review & editing:** Jens Koed Madsen.

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
