## [Decision Letter · Decision Letter 0]

26 Jul 2024

PONE-D-24-17410Goal-Line Oracles: A Longitudinal Exploration of the Wisdom of the Crowd for Football PredictionsPLOS ONE

Dear Dr. Madsen,

Thank you for submitting your manuscript to PLOS ONE. After careful consideration, we feel that it has merit but does not fully meet PLOS ONE’s publication criteria as it currently stands. Therefore, we invite you to submit a revised version of the manuscript that addresses the points raised during the review process.

In a first stage I invited two reviewers who are In a first stage I invited two reviewers who both are experts within the field you adress in your paper. While reviewer 1 identies some problems with the current version, the overall evaluation is positive. Reviewer 2 is rather critical in his/her assessment. After reading the paper myself, I decided to obtain a third review, which is now available. I would ask you to base your revision on reviewers 1 and 3, but I would still be pleased if you would address the criticism of reviewer 2. 

We look forward to receiving your revised manuscript.

Kind regards,

Prof. Dr. Florian Follert

Academic Editor

PLOS ONE

Journal Requirements:

Reviewers' comments:

Reviewer's Responses to Questions

**Comments to the Author**

1. Is the manuscript technically sound, and do the data support the conclusions?

Reviewer #1: Partly

Reviewer #2: No

Reviewer #3: Partly

2. Has the statistical analysis been performed appropriately and rigorously? 

Reviewer #1: Yes

Reviewer #2: No

Reviewer #3: Yes

3. Have the authors made all data underlying the findings in their manuscript fully available?

Reviewer #1: No

Reviewer #2: Yes

Reviewer #3: No

4. Is the manuscript presented in an intelligible fashion and written in standard English?

Reviewer #1: Yes

Reviewer #2: Yes

Reviewer #3: Yes

5. Review Comments to the Author

Reviewer #1: I thank the authors for this quality work and the editor for the invitation to review. This work explores the ability of crowds and xG to surpass betting odds to forecast football matches results. Please find my comments attached.

Reviewer #2: The fundamental weakness of this paper is the lack of discussion and explanation for the primary alternative predictor of match scores and outcomes. Moreover, the only studied referenced for that alternative is from a journal called Journal of Student Research". That is just not sufficient for a convincing or compelling academic study.

Reviewer #3: Thank you for providing me with the opportunity to review your paper.

Applying the wisdom of the crowd to football predictions can provide valuable insights.

I think the manuscript has potential for publication.

However, I have a number of comments for your attention, in particular in the results section.

The introduction presents relevant background information and reviews a fairly limited but sufficient literature to derive the two hypotheses tested.

For H1, I would suggest adding as a limitation that Wisdom of the Crowd ExG estimates can only take limited values without decimals, while observed ExG can take many more values that include decimals.

The method is straightforward but works well, although one may argue that a limited ‘crowd’ responded to the survey (15 to 85 respondents, with average around 25).

This could be added as a limitation.

Relatedly, I would expect further discussion about potential sample biases, for example the risk of having fans supporting the same team(s) in your fairly small sample of colleagues, friends and fans recruited through social media, or the potential higher influence of outliers in your sample compared to larger ones (e.g., the Arsenal fan mentioned in your discussion).

The results have potential to provide valuable insights but require further consideration.

In the Expected Goals subsection, the limitation I suggest above for H1 may partially explain your result when comparing Wisdom of the Crowd and observed ExG.

I would be interested in what happens if you round observed ExG to the nearest numbers without decimals then compare to the Wisdom of the Crowd ExG.

For the regression, I would avoid talking about correlation (otherwise just calculate the correlation coefficient and its significance), maybe relationship or association instead.

One may argue that the R2 is not particularly high (less than 20%), however it is related to the limitation I suggest for H1.

A minor detail at the start of the Expected Goals subsection: in the last but two row page 14, it should be ‘all’ instead of ‘alle’.

In the Outcomes subsection, betting agencies were not mentioned as part of the comparison in H2, please amend the hypothesis accordingly.

In addition, I would suggest applying 0.5 in both predictive heuristics about differences in ExG predictions (you should make explicit you refer to differences) instead of 0.2 in the first one; that is, I would suggest having only one predictive heuristic, otherwise further explanation about why 0.2 and not another value, and why not a single value for all situations (i.e., only one predictive heuristic) should be provided.

H2 is tested based on four individuals only.

I would suggest performing further robustness checks, for example by including individuals who participated in at least half of the rounds.

For the comparison with betting odds, why 36 and not 38 rounds?

A minor detail in the Outcomes subsection: page 19, in the sentence starting with ‘The predictions’, 36 rounds instead of matches.

The discussion and concluding remarks section is fine with me overall.

However, it is strange to refer to three hypotheses when only two were introduced earlier.

I hope you find my feedback fair and helpful.

I wish you all the best with further development of your research.

6. PLOS authors have the option to publish the peer review history of their article (what does this mean?). If published, this will include your full peer review and any attached files.

Reviewer #1: No

Reviewer #2: No

Reviewer #3: **Yes: **Nicolas Scelles

---

## [Author Response · Author response to Decision Letter 0]

9 Sep 2024

Dear Florian and anonymous reviewers

In the following, I have responded to comments from reviewers as well as from editor. I have left the comments and suggestions in black and me responses in red. 

Comments from the editor

Thank you for submitting your manuscript to PLOS ONE. After careful consideration, we feel that it has merit but does not fully meet PLOS ONE’s publication criteria as it currently stands. Therefore, we invite you to submit a revised version of the manuscript that addresses the points raised during the review process.

In a first stage I invited two reviewers who are In a first stage I invited two reviewers who both are experts within the field you address in your paper. While reviewer 1 identifies some problems with the current version, the overall evaluation is positive. Reviewer 2 is rather critical in his/her assessment. After reading the paper myself, I decided to obtain a third review, which is now available. I would ask you to base your revision on reviewers 1 and 3, but I would still be pleased if you would address the criticism of reviewer 2.

I am grateful to be able to submit a revised version of the manuscript. I hope the responses below satisfactorily addresses the constructive and helpful comments made by reviewers 1-3. As suggested, I have focussed predominantly on the comments from reviewers 1 and 3, but I have sought to engage with reviewer 2 as well, as they raise some interesting and useful points.

As per the journal’s request, I have made all data files available for OSF on the following link: https://osf.io/kg5rv/?view_only=26b46cde41df40b29fd48cb90590ec6c

Comments from reviewer 1

I. General comment

I want to thank the authors for this quality work and the editor for the invitation to review. This work explores the ability of crowds and xG to surpass betting odds to forecast football matches results. The application of wisdom of crowds to match forecasting is a promising topic, and the approach of the authors seems relevant. I am quite positive about this work and, as I will focus more on the issues in the following sections, I want to emphasize that I believe the paper will eventually make it to the publication stage. However, the current form of the paper presents several issues. I hope these comments will help the authors to improve his work. Below, I offer my comments, some critical, others as suggestions for added value. Last, I offer my notes on typos and minor issues to be addressed.

I am grateful for the careful review and constructive suggestions below. I have sought to engage with the suggestions and feedback – I believe the paper has improved considerably as a result of the additional sections and analyses that I have added to clarify and contextualise the study and results. 

If a second round takes place, I would suggest numbering the pages (and maybe the lines) to ease the review. The page numbers below refer to the pages in the proof I received. As a reference, the introduction starts on page 8.

I have added page numbers in line with this suggestion

a. Critical issues

My main concerning the manuscript is the relative weakness of the empirical setup, which limits the reach of the results. However, the research question is particularly relevant, and the results are promising. To overcome the empirical issues and especially, the limited sample sizes, I would like to see robustness checks and power/sensitivity analyses (for example, but any suggestion from the author is welcome). What is important to me is that the author shows that the setup does not influence the results: changing the aggregation heuristic, using different tests, etc.

I am grateful to the suggestions below – I agree that the empirical setup has limitations, which we have discussed in further detail (see below). I have run the suggested analyses to check robustness of model predictions. I feel this strengthens the argument in the paper considerably, as it places the WoC model in a better context. 

1. Given that the sample sizes are limited, I would like to see some power analysis to strengthen

the robustness of the results.

I agree that the power analysis was missing. To remedy this, I have included the following in the manuscript:

For each round of matches, we average the scores of each participant to generate the prediction for expected goals, which is then compared with ExG. For each round, there are ten matches (i.e., ten options for comparing predictions to observations). As there are 38 rounds, we have 380 averages in total to be compared with observations. Given the study design, we run a G*Power 3.1 (Faul et al., 2009) analyses of repeated measures (within factors) for an ANOVA to determine the sample size needed to minimise type-I and type-II errors and ensure a strong statistical power. Using effect size (f) = 0.2, significance level (α) = 0.05, power level (1-β) = 0.95, number of groups = 1, number of measurements = 38, and a nonsphericity correction = 0.5, a sample size of 21 participants is recommended. As the study averages 25.21 participants per round, it suggests that the study was adequately powered. 

2. On p.18, I would like to know how sensitive the results are to the heuristics used (0.04 gap = draw and >0.5 gap = win)? Would simply compare WoC estimates (if WoCa > WoCb then a win, with a tolerance for draws) yield the same results? Some robustness checks would be helpful to strengthen your results.

I am grateful for the suggestion to test the robustness of this, as the 0.2 heuristic for draws was set intuitively given the exploratory nature of the study. As the above suggestion would only impact draw predictions, I have included the following additional analysis to increase the robustness of the results:

As described above, we used a heuristic to assist predictions (e.g., if the predicted gap between two teams is less than 0.2, we estimated a draw). To rest the robustness of the ‘draw’ heuristic, we trial an alternative model where differences of 0.1 or less (instead of 0.2) yield a draw. That is, if Team A has an ExG of 1.23 and Team B has an ExG of 1.05, the original heuristic would predict a draw (as the difference is less than 0.18) while the new heuristic would predict that Team A would win. While the original heuristic yielded an overall prediction rate of 50% exactly, the new heuristic does slightly better and yields a prediction rate of 52.1%. This suggests that the original heuristic was set too wide and that a difference of 0.1 or less may be a better heuristic for estimating draws. Of course, as only 94 of the total 380 matches were predicted as draw, we cannot say if the differences are statistically significant over a larger sample size. Nonetheless, the results suggest that a tighter heuristic would be more successful as a prediction assistance. 

3. The authors use mean aggregation strategy to summarise the opinion of the crowd. Why not test other strategies (the one proposed by Navajas and colleagues, median, etc.)? In my opinion, this should not take to much time and computation while enhancing the robustness of the analysis. 

As with the above, I am grateful for this suggestion, as it checks the robustness of the findings. I have included the following in the manuscript: 

To gauge the robustness of the predictions, we compare predictions with other methods of gauging Wisdom of the Crowds, such as Navajas et al. (2018) or taking the median rather than the mean estimate. Navajas and colleagues (2018) provide an alternative method to generate WoC estimates. Here, participants provide individual estimates. After this, they are grouped together in small groups to discuss the same question to reach a consensus. The final collective estimate is derived by aggregating these group consensus decisions, which has been shown to be more accurate than averaging individual opinions. However, as we cannot get participants to deliberate, we can only provide a rough approximate of this method by dividing respondents into five random groups. We then take the mean of these sub-groups (representing reaching consensus within the synthetic groups) and then generate a mean by aggregating these ‘group’ consensus decisions. However, as this does not allow for deliberation (and therefore persuasion and belief revision), the aggregate eventually yield more or less the same outcomes as just averaging across all participants (comparing the Navajas approximate with simple means via a linear regression yields an R2 of 0.998, indicating that the results are the same). We therefore do not pursue the Navajas method further. 

 As an alternative to using the mean, we may query if taking the median yields a better prediction. To do so, we take the median value of each prediction to generate the estimate number of goals for each team. However, this has worse overall predictive success than the averaging model, as the median model only yields 47.6% correct guesses (compare with 50% for the 0.2 averaging model and 52.1% for the 0.1 averaging model). Branching across the three models, we observe the following rate of success: Mean estimate (0.1 draw heuristic): 52.1% correct, Mean estimate (0.2 draw heuristic): 50.0% correct, Median estimates: 47.6% percent correct. This indicates that the best WoC method for the current project is aggregate means with a draw heuristic of 0.1. 

4. On p. 16, the author presents “exploratory corrections of predictions” but I do not get the

purpose of this operation: if you remove the average difference between the two, you will end up with close mean, this is a logical consequence but does not prove your point in my opinion.

I agree with this – I have removed the section, as it does not advance any argument in the text and seems self-evident. 

5. From a conceptual perspective, xG is more about chance creation than about goal scored. On the other hand, the WoC assessment is about goal scored (given the formulation of the question asked to participants). Therefore, I think WoC assessment and xG are not directly comparable because chance creation and goal scored are two related yet different issues. Maybe comparing xG and WoC assessment relations to actual goals would be a better idea than comparing WoC assessment to xG directly.

I clarify that the predictions are compared against both Expected Goals as well as actual goals to strengthen the analyses, as this was not sufficiently clear in the original manuscript. Specifically, we have added the following to the text:

As such, we have two measures for comparison with different benefits and limitations. First, Wisdom of the Crowds predictions compared with actual goals scored benefit from the fact that goals are observable and genuine outcomes. However, given the considerable noise in football outcomes, this is subject to variance. For example, a team may be dominated by their opposition and produce very few chances and still win 1-0 on an extremely unlikely shot from far away at the end of the game. In this case, predictions may be wrong on the outcome (the dominated team winning) but right in predicting the level of domination. Second, Wisdom of the Crowd predictions compared with XG benefit from comparing expected dominance (predictions) with actual dominance (XG expressed as the number and quality of chances a team produces). However, this is limited by the fact that XG does not necessarily translate into actual outcomes. By comparing predictions to both, we get a richer picture of the Wisdom of the Crowd’s capacity to predict football matches (outcome or chance-creation). 

6. Overall, I think that the literature section lacks some important elements:

- On the concept of “wisdom of the crowds” I general, I suggest you refer to Surowiecki (2005) I have included this reference in the literature review. 

- On the forecasting of football match results with “human opinion”, you may read Brown et al. (2018) who use sentiment analysis from Twitter posts, and Peeters (2018), who use Transfermarkt crowd-sourced values. I cited Peeters previously, but I have included Brown et al. to complement the literature review. 

- On p.11 the author states “Related to the current study, other papers explore crowd predictions and transfer market valuations (e.g. 12, 13).” Research on Transfermarkt values is the main literature on wisdom of the crowds applied to football and I believe you should spend more time on it. For empirical and conceptual overview I suggest Franceschi et al. (2023a, 2023b). I have added these to the presentation of the transfermarkt values. 

- The author should spend more time on expected goals and give references (Pollard et al., 2004; Ensum et al., 2005, Stats Perform 2012). They could also spend some time discussing the limit of the xG approach (Spearman, 2018) For expected goals, I have added the following section to discuss them as a metric, including their limitations. 

Of course, there are limits to XG as an accurate measure of dominance and chance-creation. For one, it is difficult to calculate the exact probability of converting a chance to a goal given the fact that most situations are somewhat unique (e.g., distance from goal may be similar in two situations while the placement of defenders and supporting attacking players may increase or decrease the probability in different situations). Additionally, there are different ways to measure chance creation and the probability of conversion. For example, Spearman (20) uses off-the-ball metrics (e.g., a tall player who is unmarked during a corner kick).

7. On p.17, in my opinion, the predicting power of the wisdom of the crowd to predict xG is not so high in terms of R². I would like the author to critically reflects on these results. In particular the author says “This supports H1 that Wisdom of the Crowds ExG estimates will not differ from observed ExG”, but such a low R² is not sufficient to verify this hypothesis in my opinion. It is arguable however and the author should provide further details to support their point. It would be interesting to rely on metrics traditionally used to assess prediction power rather than R² (MAE, RMSE, MAPE). I would be interested to see the distribution of “residuals” (difference between WoC assessment and xG), which would be more interesting than average comparison.

I agree with this limitation to the interpretation of the results. To reflect the limited R2-value, I have included the following: 

As we see a positive correlation between predictions and XG as well as observe no significant difference between estimates and overall mean XG per game, the results provide tentative support H1 (Wisdom of the Crowds XG estimates will not differ from observed XG). These predictions positively correlate with observed XG. However, we note that the correlation is relatively weak (around 19.45% of the variance is explained by the model). Given the chaotic and dynamic nature of football, this is not surprising. As mentioned previously, matches can vary greatly in low-scoring games, as early goals may encourage otherwise dominant teams to sit back and consequently create fewer chances than expected over the course of the match. 

I have also included an MAE and RMSE and included the following

To further probe the data, we conduct Mean Absolute Error (MAE) and Root Mean Square Error (RMSE) analyses of predicted and observed XG. This yields an MAE of 0.645 and an RMSE of 0.844. As MAE uses absolute values of errors rather than squaring errors before averaging as RMSE, the former is less sensitive to outliers. Given a typical range of XG of 0-4, the MAE and RMSE analyses suggest that model predictions are reasonably close to the observed XG values. However, there is definite room for improvement, which suggests that the correlations and closeness of averages should be taken with a grain of salt.

Suggestions

8. I would suggest using the usual abbreviation for Expected Goals, i.e. xG, not ExG.

This does look cleaner – I have amended this throughout and replaced ExG with XG. 

9. The study is said to be longitudinal but my understanding is that the participants in the survey can change every week. Why is the study called longitudinal then?

I agree that this can be confusing – as a consequence, I h

---

## [Decision Letter · Decision Letter 1]

1 Oct 2024

PONE-D-24-17410R1Goal-Line Oracles: Exploring Accuracy of Wisdom of the Crowd for Football PredictionsPLOS ONE

Dear Dr. Madsen,

Thank you for submitting your manuscript to PLOS ONE. After careful consideration, we feel that it has merit but does not fully meet PLOS ONE’s publication criteria as it currently stands. Therefore, we invite you to submit a revised version of the manuscript that addresses the points raised during the review process.

I would like to inform you that Reviewer 4 has provided some additional minor comments regarding your manuscript. While these points are mostly minor, I believe that addressing them could further enhance the readability of your paper.

You are not required to implement all the suggestions, but I strongly recommend correcting any typos. Additionally, there is one point of confusion that the reviewer himself has acknowledged; I believe it could be beneficial to streamline or shorten this particular section to avoid any unnecessary ambiguity.

Please consider this a conditional acceptance of your paper, pending these final revisions. I look forward to receiving the updated version of your manuscript.

We look forward to receiving your revised manuscript.

Kind regards,

Florian Follert

Academic Editor

PLOS ONE

Journal Requirements:

Reviewers' comments:

Reviewer's Responses to Questions

**Comments to the Author**

1. If the authors have adequately addressed your comments raised in a previous round of review and you feel that this manuscript is now acceptable for publication, you may indicate that here to bypass the “Comments to the Author” section, enter your conflict of interest statement in the “Confidential to Editor” section, and submit your "Accept" recommendation.

Reviewer #3: All comments have been addressed

Reviewer #4: (No Response)

2. Is the manuscript technically sound, and do the data support the conclusions?

Reviewer #3: Yes

Reviewer #4: Yes

3. Has the statistical analysis been performed appropriately and rigorously? 

Reviewer #3: Yes

Reviewer #4: Yes

4. Have the authors made all data underlying the findings in their manuscript fully available?

Reviewer #3: No

Reviewer #4: Yes

5. Is the manuscript presented in an intelligible fashion and written in standard English?

Reviewer #3: Yes

Reviewer #4: Yes

6. Review Comments to the Author

Reviewer #3: (No Response)

Reviewer #4: Thank you to the author for their careful revision. Please find my report attached. I hope this helps.

7. PLOS authors have the option to publish the peer review history of their article (what does this mean?). If published, this will include your full peer review and any attached files.

Reviewer #3: **Yes: **Nicolas Scelles

Reviewer #4: No

---

## [Author Response · Author response to Decision Letter 1]

4 Oct 2024

Response to reviewer

I. General comment

I thank the author for their answers and for engaging wholeheartedly with the reviews. Most of my points have been addressed and I am happy with the results. My remaining desired revisions concern the readability of the paper which should be improved. Although I try to provide helpful suggestions, I encourage the author to go beyond my comments to improve the overall structure and readability.

Many thanks for the kind response and constructive comments below. I have engaged with these with responses in red. 

II. Recommendation

I would recommend minor revision to the paper.

III. Revision

On p.9 and 10, I would swap the paragraphs starting with “Second, the nature… ” and “For each round of matches… ” and remove the mention “which is sufficient statistical power as discussed below” as it would be discussed right after.

I have followed this suggestion. 

On p.10, the conclusion to the power analysis is: “sample size of 21 participants is recommended. As the study averages 25.21 participants per round, it suggests that the study was adequately powered.”. I believe the author takes a shortcut here: the average may well be 25.21 with some rounds having less than 21 participants, hence being underpowered. As a matter of fact, I checked 3 rounds on OSF (rounds 1, 10 and 12) and found out that the round 12 file contains only 19 rows. In my opinion, this is not a big issue, but I would appreciate a better formulation and more transparency (providing an appendix with the number of participants per round would be great).

I agree with the reviewer on this point. I have added the following to clarify the concern regarding underpowered rounds: 

Nonetheless, some rounds have fewer than required participants, making them more vulnerable to variation or biases from individual respondents. However, as discussed below, when looking at the percentage of correct guesses in weeks with fewer than 20 participants, there are no differences compared with rounds that are adequately powered. Therefore, with some hesitation concerning some rounds, the study appears reasonably powered. The results should nonetheless be seen in this light and future research should replicate the study with more respondents for all rounds.

I hope this provides the transparency more clearly. 

On p.16, based on my suggestion, the author details Navajas et al.’s method to derive WoC estimates. I apologize as I understand only now that it is not feasible in the article’s empirical context. Therefore, it is not necessary to discuss the method in such details, and I believe that the author should shorten the paragraph and ease the reading. I sincerely apologize for the inconvenience. On the other hand, the section concerning the median is a great addition.

It was good to engage with the method regardless so I am happy to have done so. I have shortened the paragraph in line with suggestions. 

Overall, the author should enhance the readability by clarifying the structure of the paper. Here is a non-exhaustive list of ideas:

- better balance the sections (5 pages for introduction, 2.5 pages for methods, 7.5 for results and 2 for discussion and conclusion) by adding a literature review section which would shorten the introduction and make it more impactful;

- provide an outline at the end of the introduction ;

- develop sharper subtitles in the Results section.

I have divided the manuscript further with subtitles in the results section and a clear literature review section. I hope this provides a better balance through sign-posting. 

IV. Typos and corrections

On p.13, I think a word is missing before H1: “the results provide tentative support H1 (Wisdom of the Crowds XG estimates will not differ from observed XG).”

I have corrected this and gone through types throughout the manuscript. I hope the revisions are appropriate.

---

## [Editor Report · Decision Letter 2]

8 Oct 2024

Goal-Line Oracles: Exploring Accuracy of Wisdom of the Crowd for Football Predictions

PONE-D-24-17410R2

Dear Dr. Madsen,

We’re pleased to inform you that your manuscript has been judged scientifically suitable for publication and will be formally accepted for publication once it meets all outstanding technical requirements.

Kind regards,

Florian Follert

Academic Editor

PLOS ONE

---

## [Editor Report · Acceptance letter]

18 Oct 2024

PONE-D-24-17410R2 

PLOS ONE

Dear Dr. Madsen, 

I'm pleased to inform you that your manuscript has been deemed suitable for publication in PLOS ONE. Congratulations! Your manuscript is now being handed over to our production team.

Kind regards, 

on behalf of

Prof. Dr. Florian Follert 

Academic Editor

PLOS ONE